# Discovery of lipid binding sites in a ligand-gated ion channel by integrating simulations and cryo-EM

**Cathrine Bergh[1], Urška Rovšnik[2], Rebecca Howard[1,2], Erik Lindahl[1,2]\***

[1]Science for Life Laboratory & Swedish e-Science Research Center, Department of Applied Physics, KTH Royal Institute of Technology, Stockholm, Sweden; [2]Science for Life Laboratory, Department of Biochemistry and Biophysics, Stockholm University, Stockholm, Sweden

**Abstract** Ligand-gated ion channels transduce electrochemical signals in neurons and other excitable cells. Aside from canonical ligands, phospholipids are thought to bind specifically to the transmembrane domain of several ion channels. However, structural details of such lipid contacts remain elusive, partly due to limited resolution of these regions in experimental structures. Here, we discovered multiple lipid interactions in the channel GLIC by integrating cryo-electron microscopy and large-scale molecular simulations. We identified 25 bound lipids in the GLIC closed state, a conformation where none, to our knowledge, were previously known. Three lipids were associated with each subunit in the inner leaflet, including a buried interaction disrupted in mutant simulations. In the outer leaflet, two intrasubunit sites were evident in both closed and open states, while a putative intersubunit site was preferred in open-state simulations. This work offers molecular details of GLIC-lipid contacts particularly in the ill-characterized closed state, testable hypotheses for state-dependent binding, and a multidisciplinary strategy for modeling protein-lipid interactions.

**\*For correspondence:**
erik@kth.se

**Competing interest:** The authors declare that no competing interests exist.

## eLife assessment

The authors use a combination of structural and MD simulation approaches to characterize phospholipid interactions with the pentameric ligand-gated ion channel, GLIC. The general agreement between structures and simulations increases confidence in the description of the lipid interaction poses and provides a **solid** basis for the prediction of a state-dependent interaction site where lipids could dynamically modulate channel gating. The results will be very **useful** to understand the nature of phospholipid interactions with pentameric ligand-gated ion channels, although the functional or structural significance of these lipid interactions remains to be verified.

## Introduction

Pentameric ligand-gated ion channels (pLGICs) can be found in numerous cell types, particularly in the postsynaptic membrane of the vertebrate nervous system, where they transduce electrochemical signals upon neurotransmitter binding (*Tasneem et al., 2005*). Gating in pLGICs is initiated by ligand binding in the extracellular domain (ECD), which is believed to lead to large twisting and blooming motions in the ECD, followed by rearrangements in the transmembrane domain (TMD), which eventually lead to opening of the central transmembrane pore. In addition to soluble ligands and numerous allosteric modulators, lipid components of the plasma membrane have been shown to bind to a variety of membrane proteins, potentially modifying their stability or gating (*Duncan et al., 2020*). For pLGICs, lipid composition was shown to influence reconstitution of functional activity in the *Torpedo*

nicotinic acetylcholine receptor (nAChR) as early as the 1980s (*Popot et al., 1981*; *Criado et al., 1982*; *Fong and McNamee, 1986*; *daCosta et al., 2002*). Multiple members of the pLGIC family, particularly nAChRs and prokaryotic channels cloned from *Gloeobacter violaceus* (GLIC) and *Dickeya dadantii* (formerly *Erwinia chrysanthemi*, reflected in the channel name ELIC), have since demonstrated functional sensitivity or co-purification with membrane lipids (*Thompson and Baenziger, 2020*). It has been suggested that lipids interact with membrane-facing loops from the ECD, or with the four membrane-spanning helices (M1-M4) of the TMD. However, many structural details of these lipid interactions remain poorly understood (*Poveda et al., 2017*).

A critical challenge to elucidating lipid-protein interactions is their typically poor definition in experimental densities. Among more than 100 deposited structures in the protein data bank for GLIC crystallized under activating conditions, several lipid poses have been reported (*Zhu et al., 2018*; *Masiulis et al., 2019*; *Laverty et al., 2019*; *Walsh et al., 2018*; *Gharpure et al., 2019*; *Rahman et al., 2020*; *Zhao et al., 2021*). These include an outer-leaflet site on the complementary face of a single GLIC subunit, an inner-leaflet site on the principal face of a subunit, and an interfacial inner-leaflet site making contacts with two neighboring subunits (*Thompson and Baenziger, 2020*). In one case, an additional outer-leaflet density was built as a detergent molecule (*Hu et al., 2018*). However, the positions and orientations of these membrane components or mimetics vary between structures and typically include only a subset of their constituent atoms.

Another issue is the potential dependence of lipid interactions on the functional state of the membrane protein. In most members of the pLGIC family, binding of a chemical stimulus to the ECD favors a transition from an unliganded, nonconducting state to a liganded, conducting state, and in some cases to a subsequent liganded, desensitized state. Although terminology in the field may vary, conditions predicted to favor the first versus second of these states are referred to here as *resting* versus *activating*; the corresponding conformations of the TMD pore are designated *closed* versus *open*. No lipids were reported with the lone X-ray structure of GLIC under resting conditions, raising the possibility that interactions could be state-dependent, or obscured by the relatively low overall resolution of this dataset (4.0 Å) (*Sauguet et al., 2014*). Independent validation of phospholipid interactions in the closed as well as open states of GLIC would provide a valuable baseline for investigating such contacts in the larger family of pLGICs, including pharmacologically important effects of components such as cholesterol and neurosteroids that modulate LGICs (*Thompson and Baenziger, 2020*; *Kim and Hibbs, 2021*).

Molecular dynamics (MD) simulations offer an alternative approach to visualizing lipid-protein interactions. All-atom simulations of membrane proteins typically involve inserting an experimental structure in a simplified lipid bilayer surrounded by water and ions, all of which move freely according to physical principles encoded in the force field. Simulations on both atomistic and coarse-grained levels have recently enabled the identification of specific lipid interactions in pLGICs such as ELIC (*Sridhar et al., 2021*; *Dietzen et al., 2022*), nAChRs (*Sharp and Brannigan, 2021*), glycine (*Dämgen and Biggin, 2021*), and serotonin-3 receptors (*Crnjar and Molteni, 2021*), but were typically limited to shorter timescales or specific protein conformations. We recently reported free energy landscapes for GLIC gating using Markov state modeling (MSM) of 120 μs unrestrained MD simulations (*Bergh et al., 2021*). Because GLIC is activated by external acidification (pH < 6), resting versus activating conditions could be approximated throughout these simulations by setting titratable amino acid residues as deprotonated versus protonated, respectively. Under these two conditions, different distributions of functional states were captured, consistent with the relative stabilization of open versus closed channels upon activation. Although our previous analyses focused on conformational changes of the protein alone, the same simulation data encodes a wealth of additional features, including membrane lipids. We also recently reported the first structural data for GLIC using cryogenic electron microscopy (cryo-EM), which enabled the reconstruction of multiple new closed structures (*Rovšnik et al., 2021*).

Here, we present the first cryo-EM structure of GLIC in a closed conformation with multiple bound lipids resolved in each subunit. These were built into non-protein densities evident in the cryo-EM data, further informed by closed-state lipid occupancies independently identified and characterized from extensive MD simulations. Five distinct closed-state lipid-interaction sites per subunit covered intra- and intersubunit cavities in the inner and outer leaflets. We also compared open-state lipid occupancies from our simulations with previous open X-ray structures, identifying six possible poses. These analyses allowed prediction of novel lipid interaction features in GLIC, including a potentially

state-dependent binding site at the outer subunit interface, and a potential role for lipid-tail saturation in binding at the inner leaflet, further validated through mutant simulations. In addition to casting light on selective lipid interactions with LGICs, this work demonstrates the combined power of experimental densities and molecular simulations to support the characterization of lipid binding sites in systems challenged by heterogeneous or otherwise low-resolution data.

## Results

### MSM and cryo-EM data resolve lipid-protein interactions

We previously reported cryo-EM structures of GLIC in multiple closed states (*Rovšnik et al., 2021*) and demonstrated how MSM can predict shifts in the GLIC gating landscape under resting versus activating conditions (*Bergh et al., 2021*). In this work, we have combined new cryo-EM reconstructions with simulations newly extended to achieve better sampling of lipids, which made it possible to identify five previously uncharacterized lipid poses interacting with each subunit in the closed state of GLIC (*Figure 1A*).

For computational quantification of lipid interactions and binding sites, we used molecular simulations of GLIC conducted under either resting or activating conditions (*Bergh et al., 2021*). As described in Materials and methods, resting conditions corresponded to neutral pH with most acidic residues deprotonated; activating conditions corresponded to acidic pH with several acidic residues protonated. Both open and closed conformations were present in both conditions, albeit with different probabilities. Simulation frames previously identified as either closed or open (*Figure 1B*) were clustered to their corresponding state, and used to generate densities occupied by lipids that contacted the protein in at least 40% of simulation frames (*Figure 1D and F*). Although the stochastic nature of simulations resulted in nonidentical lipid densities associated with the five GLIC subunits, patterns of lipid association were notably symmetric (*Figure 1—figure supplement 1*). In order to distinguish presumed functional endpoints, snapshots classified into macrostates other than closed or open (*Figure 1B*, colored boundaries) were excluded from occupancy calculations, except as indicated below. In parallel, mean duration times of lipid contacts were quantified for each amino acid residue in each 1.7 µs trajectory (50 seeds × 2 conditions × 5 subunits=500 trajectories; *Figure 1C*). In some cases, we validated notable contacts by additional simulations in the presence of targeted mutations (*Figure 1E*).

In our cryo-EM work, a new GLIC reconstruction was generated by merging previously reported datasets collected at pH 7, 5, and 3 (*Rovšnik et al., 2021*). The predominant class from the merged data corresponded to an apparently closed channel at an overall resolution of 2.9 Å, the highest resolution yet reported for GLIC in this state (*Figure 1—figure supplement 2*, *Table 1*). After building all residues, additional non-protein densities were evident in both the outer and inner leaflets of the predicted membrane region. These were further sharpened by post-processing with ResolveCryoEM (*Terwilliger et al., 2020*) and LocScale (*Jakobi et al., 2017*). From the combined observations of these cryo-EM densities (*Figure 1G*) and occupied densities in closed-state simulations (*Figure 1F*), five lipids were built in association with each of the five GLIC subunits, for a total of 25 lipid molecules (*Figure 1H*, *Figure 1—figure supplement 1C and D*).

Lipids built in this structure formed close contacts with GLIC residues in several membrane-proximal regions. These included an ECD motif with a characteristic proline/cysteine-rich turn, known as the Pro-loop (or Cys-loop in eukaryotes); N-terminal residues of the M1 helix; regions in and surrounding the M2-M3 loop; interfacial residues at the C-terminal end of M3; and inward- and outward-facing regions of M4 (*Figure 2A*). In several cases, these regions corresponded to longer-duration lipid interactions in molecular simulations (*Figure 2B and C*). In the inner leaflet, relatively long-lasting interactions were found at the subunit interface, involving buried residues in both the M1 and M3 helices. In the outer leaflet, longer interactions were focused around the pre-M1 and M2-M3 loops. Lateral lipid diffusion coefficients were estimated to 1.47 nm²/µs for bulk lipids and 0.68 nm²/µs for lipids of the first lipid shell (*Figure 2—figure supplement 1A*), which is relatively slow compared to the timescales of each trajectory (1.7 µs). However, multiple residues throughout the M1, M3, and M4 helices exchanged contacts with two to four different lipid molecules in individual simulations (*Figure 2C*). Furthermore, 1.7 µs root mean square displacement of lipids originally in the first lipid shell was 2.15 nm, and 3.16 nm in the bulk bilayer, indicating such exchanges are not limited to nearby lipids

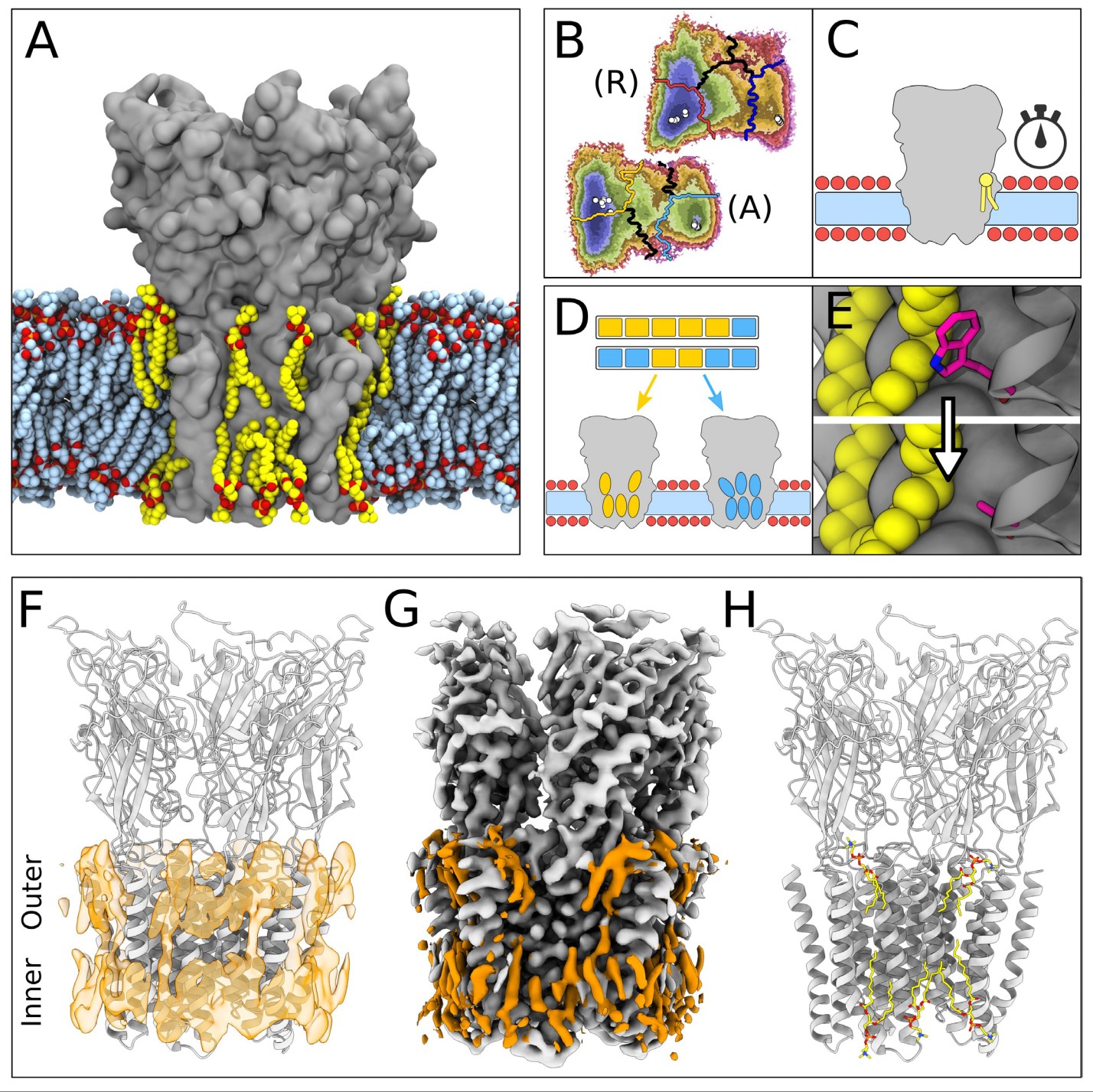

**Figure 1.** Lipid-protein interactions mapped by Markov state modeling (MSM) and cryo-electron microscopy (cryo-EM). (**A**) Overview of the molecular dynamics (MD) simulation system with GLIC (gray surface) embedded in a lipid bilayer (light blue carbon atoms) with interacting lipids highlighted (yellow carbon atoms). Lipid oxygens are shown in red; phosphorus, orange; nitrogen, dark blue. Water molecules and ions have been omitted for clarity. (**B**) Markov state models were used to cluster simulations conducted under resting (**R**) or activating (**A**) conditions into five states, including closed (left of the light or dark orange lines) and open (right of the light or dark blue lines). Black lines mark edges of other state clusters derived from MSM eigenvectors. Experimental structures are highlighted as white circles. (**C**) To characterize the stability of lipid interactions, duration times were measured in individual, unclustered, trajectories. (**D**) To identify lipid binding sites, lipid-occupied densities were obtained from state-clustered frames for closed (orange) and open (blue) states. (**E**) Further simulations were performed to test the role of residues involved in extensive lipid interactions through single residue mutations around lipid binding sites. (**F**) View of a manually built cryo-EM model of closed GLIC (gray) with computationally derived lipid occupancies in orange (semi-transparent). (**G**) Cryo-EM reconstruction of closed GLIC (gray), with partly defined non-protein densities (orange).

*Figure 1 continued on next page*

*Figure 1 continued*

(**H**) Equivalent view of a manually built cryo-EM model as in F, with newly built lipids shown as sticks (yellow, heteroatom; phosphorus, orange; oxygen, red; nitrogen, blue).

The online version of this article includes the following figure supplement(s) for figure 1:

**Figure supplement 1.** Occupational lipid densities from simulations for all five subunits and overalys with built cryogenic electron microscopy (cryo-EM) lipids.

**Figure supplement 2.** Cryogenic electron microscopy (cryo-EM) data and processing pipeline.

**Table 1.** Cryogenic electron microscopy (cryo-EM) data collection and model refinement statistics.

| Data collection | |
|---|---|
| Microscope | FEI Titan Krios |
| Magnification | 165,000 |
| Voltage (kV) | 300 |
| Electron exposure (e$^-$/Å$^2$) | ~40 |
| Defocus range (μm) | –2.6 to –3.8 |
| Pixel size (Å) | 0.82 |
| Symmetry imposed | C5 |
| Number of images | ~18,982 |
| Particles picked | ~2.7 million |
| Particles refined | 16,586 |
| *Refinement* | |
| Resolution (Å) | 2.9 |
| FSC threshold | 0.143 |
| Map sharpening B-factor | –80 |
| Model composition | |
| Non-hydrogen protein atoms | 12,010 |
| Protein residues | 1555 |
| Ligands | 0 |
| B-factor (Å$^2$) | 144 |
| RMSD | |
| Bond lengths (Å) | 0.004 |
| Bond angles (°) | 0.584 |
| Validation | |
| Molprobity score | 1.84 |
| Clashscore | 6.82 |
| Poor rotamers (%) | 0 |
| Ramachandran plot | |
| Favored (%) | 92.6 |
| Allowed (%) | 7.4 |
| Outliers (%) | 0 |

(*Figure 2—figure supplement 1B*). Thus, exchange events and diffusion estimates indicate that the duration of lipid contacts observed in this work can be at least partly attributed to interaction stabilities and not solely to sampling limitations.

## Protein and lipid determinants of state-independent inner-leaflet binding

In the inner leaflet, occupied densities from closed-state simulation snapshots were largely superimposable with those from open snapshots (*Figure 3A*, *Figure 1—figure supplement 1*), indicating lipid interactions in this region were not notably state-dependent. In both states, three types of inner-leaflet computational densities could be observed; accordingly, our cryo-EM reconstruction included three types of forked densities, which could be built as three independent lipids per subunit (*Figure 3B*). One density spanned the principal M3 and complementary M1 helices of neighboring subunits; one was located between M3 and M4 on the principal face of each subunit; and another spanned M1 and M4 on the complementary face of each subunit. Interfacial and principal-face inner-leaflet lipids have been resolved in previous open-state X-ray structures of GLIC (*Bocquet et al., 2009*) as well as other pLGIC structures (*Thompson and Baenziger, 2020*), while the complementary inner-leaflet GLIC site, to our knowledge, has not been reported before.

The lipid occupying the subunit interface accounted for some of the longest-lived interactions in the simulations, both with M1 and M3 (*Figure 3C and F*). A particularly frequent contact at this site in our simulations was the unsaturated C=C bond of 1-palmitoyl-2-oleoyl-*sn*-glycero-3-phosphocholine (POPC) (*Figure 3D and E*). Interestingly, lipids adopted two types of poses in this site, depending on the placement of the unsaturated tail. The double bond was frequently wedged between M3-T274 and M1-W217 at the intersubunit cleft, with the remainder of the lipid projecting out toward the membrane (*Figure 3D*, left). When the double bond exited the pocket, the more flexible saturated tail was able to enter

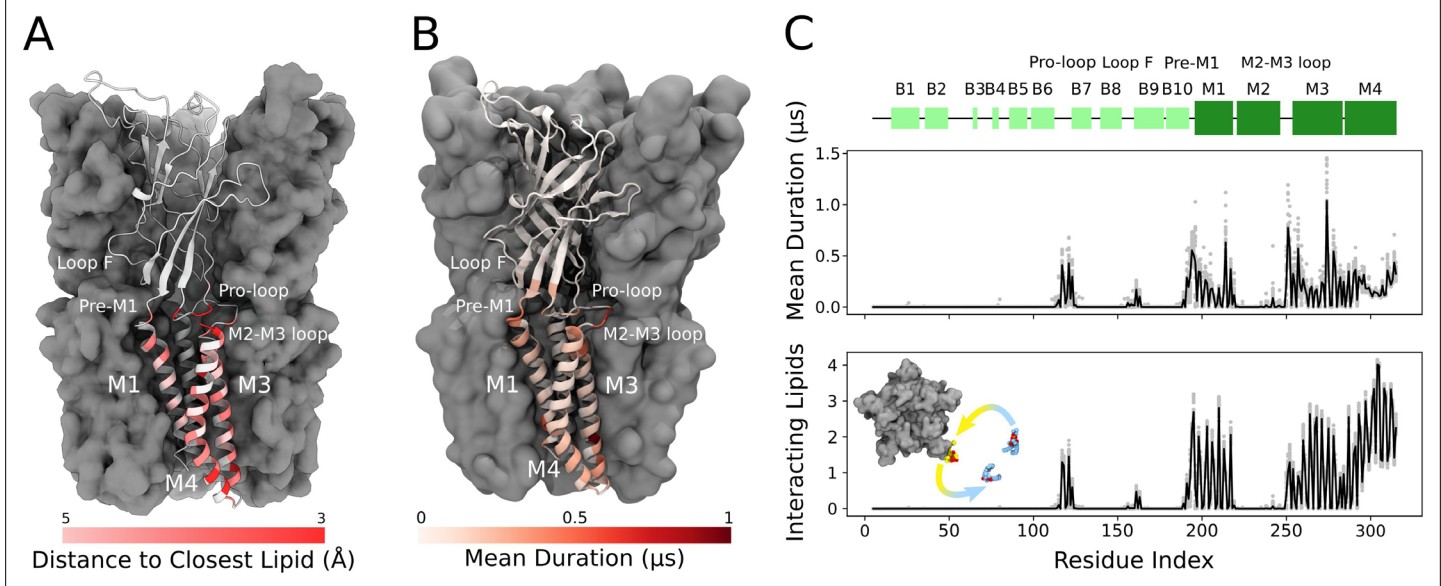

**Figure 2.** Cryogenic electron microscopy (cryo-EM) and simulations reveal lipid interaction sites. (**A**) Cryo-EM reconstruction of closed GLIC with distance to the interacting lipids colored according to the bar below. (**B**) Mean duration time of lipid-residue interactions, scaled (white-red) according to the color bar below, projected onto the closed-state GLIC structure (PDB ID: 4NPQ). For both panels, a single subunit is displayed in cartoon. (**C**) Secondary structure schematic for GLIC (top), mean duration time of each lipid-residue interaction (middle), and number of lipids interacting with each residue during simulations (bottom). Light gray points represent numbers from individual trajectories and black lines ensemble averages (500 trajectories).

The online version of this article includes the following figure supplement(s) for figure 2:

**Figure supplement 1.** Lateral diffusion of lipids initially with and without contact to the protein surface.

the intersubunit cleft and sample a range of poses characterized by contacts distributed more broadly across the lipid tail (*Figure 3E*), resulting in some deeply buried poses (*Figure 3D*, right).

To further explore the influence of specific amino acid contacts on this interfacial lipid pose, we performed additional MD simulations with mutations in targeted sites. Interestingly, the buried M3 position 274 remained the contact of longest duration, even when mutated from threonine to a smaller (alanine) or larger (tryptophan) residue (*Figure 3—figure supplement 1*). Conversely, removing the bulky tryptophan sidechain at position 217 (W217A) dramatically shortened lipid contacts at the more deeply buried T274 (*Figure 3F*). These results confirm the importance of specific sidechain identities on the protein surface (W217) in influencing lipid binding pose, whereas even long-lived contacts (T274) may be sustained more by their buried position than sidechain size.

## Intrasubunit lipids and state dependence in the outer leaflet

In the outer leaflet, closed-state cryo-EM densities enabled confident building of two lipid poses, associated with either the principal or complementary face of each subunit (*Figure 4A*, left). Lipids have previously been resolved in open and inhibited X-ray structures of GLIC at a complementary-face outer-leaflet site between the M1 and M4 helices (*Bocquet et al., 2009*; *Nury et al., 2011*; *Basak et al., 2017*; *Hu et al., 2018*, *Figure 4A*, right), comparable to that observed here in the cryo-EM closed state. To our knowledge, no lipid has been reported at any other outer-leaflet site in GLIC, though a principal-face density was built as a detergent molecule in at least one open-state X-ray structure (*Hu et al., 2018*, *Figure 4A*, right).

In closed-state simulation snapshots, outer-leaflet densities largely corresponded to our closed-state cryo-EM data, including distinct interaction sites on the complementary and principal faces (*Figure 4B*, left, *Figure 1—figure supplement 1C*). In these and other sites, occupancies and distributions were largely superimposable when calculated from Markov state models constructed independently under resting versus activating conditions (*Figure 4B and C*, *Figure 1—figure supplement 1*), indicating reproducibility of the approach and relative insensitivity of the lipid-interaction landscape

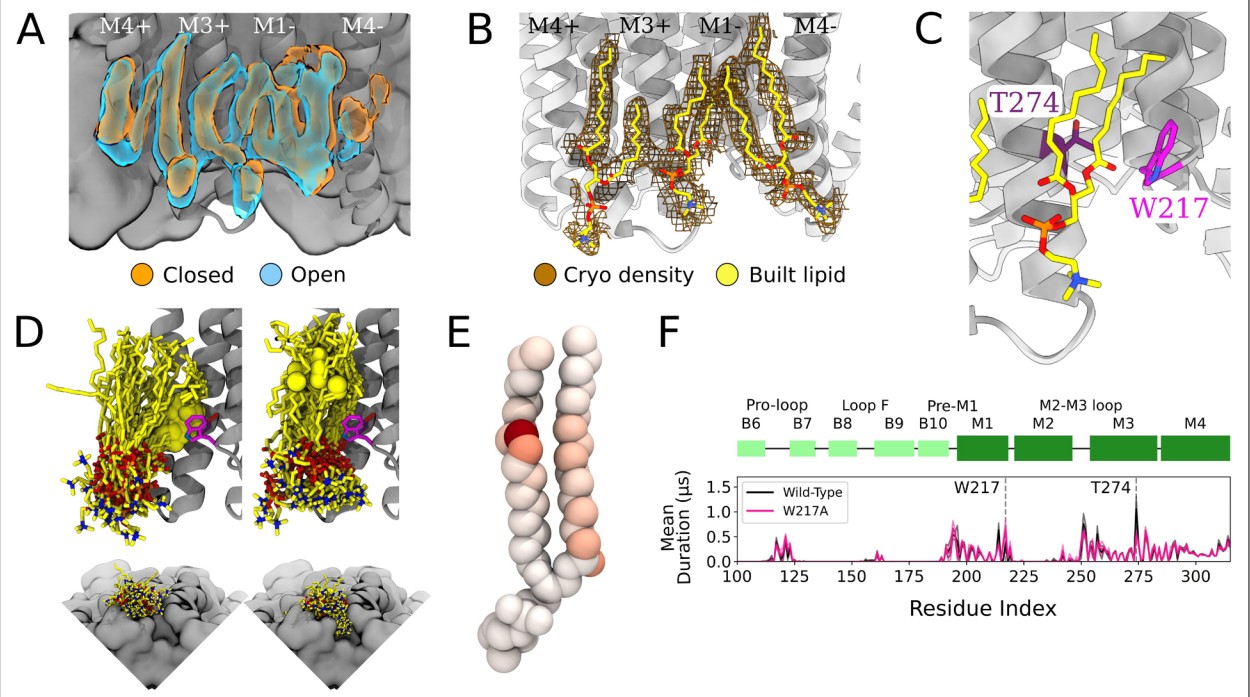

**Figure 3.** Protein and lipid determinants of state-independent inner-leaflet binding. (**A**) Computationally derived densities from the open (blue) and closed (orange) states generally agree. (**B**) Modeled lower leaflet lipids (yellow, stick, heteroatom coloring) from cryogenic electron microscopy (cryo-EM) densities (brown mesh). (**C**) Zoom-in view of the lower leaflet buried lipid interaction site at the M1-M3 subunit interface. Lipids particularly interact with residues T274 (purple) and W217 (magenta). (**D**) The 100 simulation frames that best fit the computational densities clustered into two distinct binding poses at the T274 binding site. When the tail with a double bond (yellow sphere) is situated in the pocket lipid heads are directed out from the channel (left panel), while a larger variety of poses are sampled when the lipid tail without a double bond occupies the site, allowing the head to enter into a crevice on the bottom of the channel, approaching the pore (right panel). The M4 helix is not shown for clarity in the top panels. (**E**) Number of contacts made by specific 1-palmitoyl-2-oleoyl-sn-glycero-3-phosphocholine (POPC) lipid atoms with residue T274 in simulations at both resting and activating conditions (colors span 0 contacts, white, to 137,015 contacts, dark red). For the tail with an unsaturation, interactions are concentrated around the double-bond region while the saturated tail displays interactions interspersed along the tail. (**F**) Mutation of residue W217, lining this pocket, reveals shortened interactions at the T274 binding site (magenta).

The online version of this article includes the following figure supplement(s) for figure 3:

**Figure supplement 1.** W217 and T274 mutants mean duration times compared to wild type.

to pH. However, comparing open- versus closed-state occupancies suggested state-dependent differences in lipid interactions, as described below (***Figure 4B***).

At the complementary face, lipid tails were more likely to enter a buried region of the intrasubunit pocket in the open state (***Figure 4—figure supplement 1***). Although the difference was subtle, the distribution of outer-leaflet lipids shifted in snapshots of open compared to closed conformations, increasing (5–10%) the probability of lipids penetrating 18 Å from the pore axis (***Figure 4C***). Interestingly, lipid tails entered even deeper (15 Å from the pore) in an alternative state within the GLIC gating landscape (***Figure 4—figure supplement 2A***), previously associated with channel desensitization due to relative contraction at the intracellular gate (***Figure 4—figure supplement 2B***, ***Bergh et al., 2021***). Although the lipid tails were too poorly resolved in our structural data to definitively capture buried interactions at the complementary site, it is interesting that the simulations implicated them in channel opening (and possibly in transitions further along the activation pathway) given that the deeper parts of this site have been shown to bind allosteric modulators such as propofol (***Heusser et al., 2018***).

At the principal face, interactions with lipid heads were more diverse than those with lipid tails. In our closed cryo-EM and a previously reported open X-ray structure (***Hu et al., 2018***), the lipid or detergent head in this site tilted away from the pore axis, toward M4 (***Figure 4A***). On initial inspection, lipid-head densities appeared less tilted in closed- versus open-state simulation frames (***Figure 4—figure supplement 2C***). However, lipid heads in this site were relatively flexible and did align with cryo-EM densities in several closed-state snapshots (***Figure 4D***). In fact, lowering the occupancy threshold

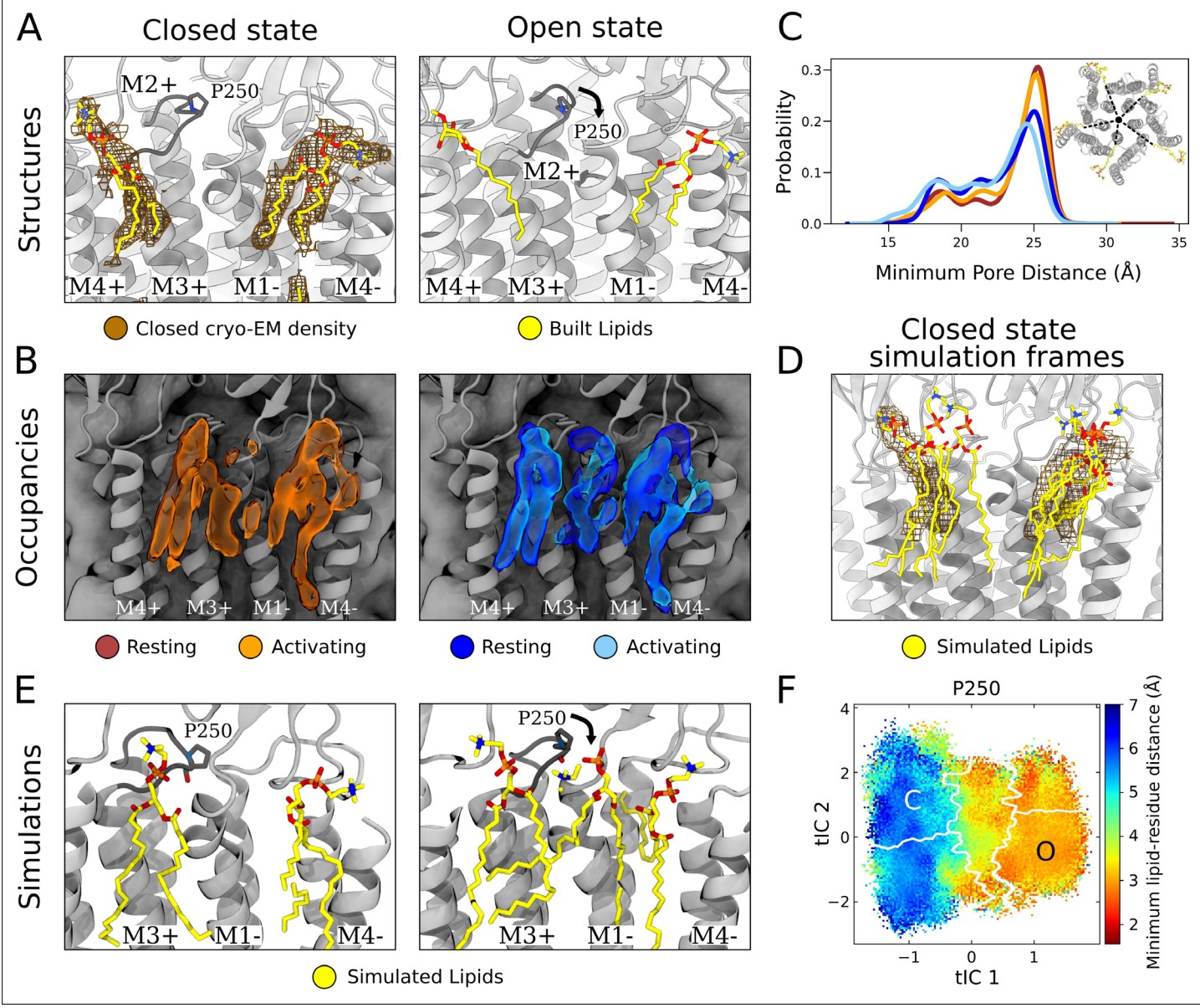

**Figure 4.** State-dependent protein-lipid interactions in the upper leaflet. (**A**) Zoom view of the outer transmembrane domain of the closed GLIC cryogenic electron microscopy (cryo-EM) structure (left, gray), showing non-protein cryo-EM densities (brown mesh) overlaid with built lipids (yellow). For comparison, the same region is shown of an open-state X-ray structure (PDB ID: 6HZW) (*Hu et al., 2018*) with built lipid and detergent in equivalent positions (right). Conformational changes of the M2-M3 loop, including residue P250, in channel opening are highlighted. Lipids are colored by heteroatom (phosphorus, orange; oxygen, red; nitrogen, blue). (**B**) Same region as in (**A**) showing densities occupied by lipids (>40%) in simulation frames clustered as closed (left, orange) or open (right, blue) states (top). Open-state differences include an enhanced density on the right-hand side of the intersubunit cleft, and a more deeply penetrating density in the intrasubunit cleft (*Figure 4—figure supplement 1*). Overlaid densities for each state represent simulations conducted under resting (dark shades) or activating (light shades) conditions, which were largely superimposable within each state. (**C**) Radial distribution of the lipid atoms closest to the upper pore, showing closer association of lipids in the open (light blue, dark blue) versus closed (orange, red) states. (**D**) Same region and cryo-EM densities (mesh) as in (**A**) overlaid with lipids from a few simulation frames where the lipids had the highest correlation with the closed-state computational occupancies in (**B**). (**E**) Snapshots from simulation frames that had the highest correlation with the occupancies (activating conditions) in (**B**). The open states are characterized by an additional lipid at the intersubunit site, interacting with the oxygen of P250. (**F**) The minimum distance between P250 and the closest lipid projected onto the free energy landscapes obtained from Markov state modeling. P250-lipid interactions are possible in the open state, but not in the closed state.

The online version of this article includes the following video and figure supplement(s) for figure 4:

**Figure supplement 1.** Alternative views of the computationally derived lipid densities at activating conditions.

**Figure supplement 2.** Comparisons of state-dependent lipid interactions in the outer transmembrane domain.

*Figure 4 continued on next page*

*Figure 4 continued*

**Figure supplement 3.** State-dependent lipid interactions and conformational changes in the upper transmembrane domain.

**Figure 4—video 1.** Morph between open and closed conformations showcasing changes in lipid interactions of the outer leaflet.

https://elifesciences.org/articles/86016/figures#fig4video1

(from 40% to 30%) revealed an alternative head-group density, tilted toward M4 and covering most of the lipid built in our cryo-EM structure (*Figure 4—figure supplement 2D*). Thus, despite an apparent discrepancy between highly populated densities in simulations and cryo-EM data, both methods supported at least partial occupancy in the built pose. Although this lipid head was primarily associated with M3-M4 on the principal face of one subunit, it also made contacts with residues on the complementary M1 helix (Q193, S196) in closed- but not open-state simulation frames (*Figure 4—figure supplement 3A, B, and C*), likely associated with conformational changes during gating.

In addition to the intrasubunit sites described above, a third outer-leaflet lipid density was observed in open-state simulation frames, wedged between two subunits (*Figure 4B*, right). This intersubunit pose was not apparent in closed-state simulations (*Figure 4B*, left), nor in closed or open experimental GLIC structures (*Figure 4A*). The lipid head made particularly close contacts with residue P250 on the M2-M3 loop, which undergoes substantial conformational change away from the pore upon channel opening, along with outer-leaflet regions of M1-M3 (*Figure 4E*, *Figure 4—figure supplement 3A, B, and C*, *Figure 4—video 1*). These conformational changes were accompanied by a flip of M1 residue F195, which blocked the site in the closed state but rotated inward to allow closer lipid interactions in the open state (*Figure 4—figure supplement 3C*, *Figure 4—video 1*). Indeed, P250 was predominantly located within 3 Å of the nearest lipid atom in open- but not closed-state frames (*Figure 4F*). Despite being restricted to the open state, interactions with P250 were among the longest duration in all simulations (*Figure 2C*) and as these binding events preceded pore opening, it is plausible to infer a role for this state-dependent lipid interaction in the gating process (*Figure 4—figure supplement 3D*).

## Discussion

Lipid interactions have increasingly been shown to play important roles in stabilizing and regulating membrane proteins (*Phillips et al., 2009*; *Cordero-Morales and Vásquez, 2018*). Particularly since the emergence of cryo-EM, a growing number of ion channel structures have been reported with detergents or lipids in membrane-facing regions (*Thompson and Baenziger, 2020*; *Levental and Lyman, 2023*). However, the resolution of bound lipids is typically low compared to the protein, making it challenging to define precise poses or interactions. As a result, lipids and detergents are often built with truncated heads, modeled as isolated hydrocarbon chains, or disregarded altogether to avoid overfitting. Here, we present the building of lipids in the closed state of GLIC by comparing non-protein densities in a relatively high-resolution cryo-EM map with lipid interactions sampled in MD simulations.

Our 2.9 Å cryo-EM structure represents, to our knowledge, the highest-resolution closed state of wild-type GLIC yet reported, and the first in which lipids could be confidently built. Notably, the GLIC sample was solubilized in detergent, which is typically thought to replace biological lipids. Nonetheless, the forked shape of non-protein densities associated with each channel subunit indicated the presence of phospholipids, evidently bound tightly enough to resist substitution by detergent. Similarly persistent lipids have been reported in at least three sites per subunit of open-state GLIC; here, we find evidence for five lipids per subunit in the closed state (*Figure 5A*). Continued improvements in membrane-protein cryo-EM may reveal the full extent of persistent lipid binding at key transmembrane sites in the larger pLGIC family.

MD simulations have recently been used to identify lipid binding sites corresponding to experimental data for both prokaryotic and eukaryotic ligand-gated ion channels (*Sridhar et al., 2021*; *Dietzen et al., 2022*), including nAChRs (*Sharp and Brannigan, 2021*), glycine (*Dämgen and Biggin, 2021*), and serotonin-3A receptors (*Crnjar and Molteni, 2021*). In most cases, the slow rate of lateral lipid diffusion has resulted in the preferential use of coarse-grained force fields able to sample longer timescales, but that may overestimate the strength of interactions (*Lamprakis et al., 2021*). In addition, such studies typically apply harmonic restraints to the protein, potentially obscuring contributions

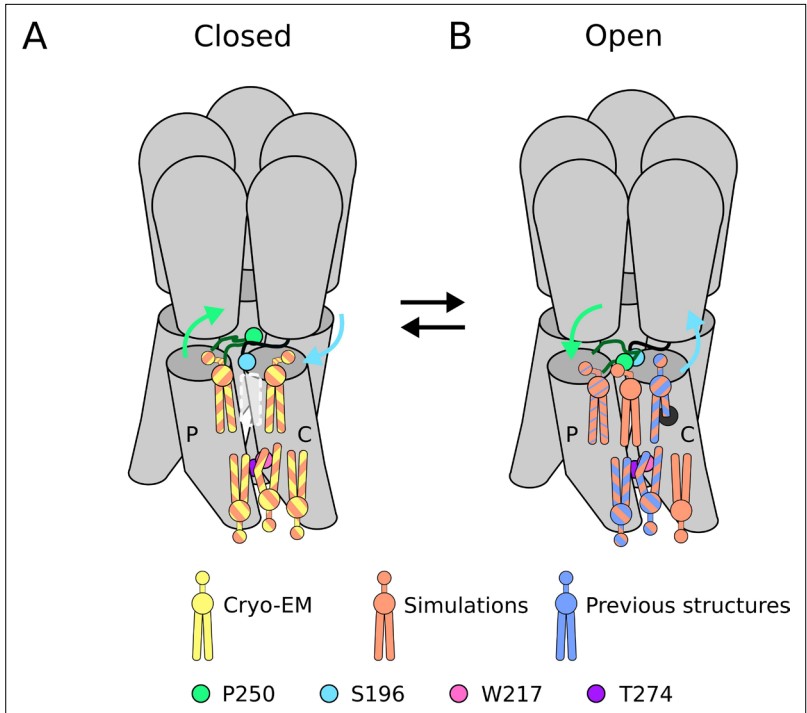

**Figure 5.** Lipid sites identified by cryogenic electron microscopy (cryo-EM) and simulations. (**A**) In the closed state, five lipid interaction sites were independently identified through both cryo-EM and simulations. In the outer leaflet, two intrasubunit sites are situated at the principal (**P**) and complementary (**C**) subunits, with the lipid at the principal subunit site interacting with S196 of the pre-M1 loop. Particularly, no lipid site could be found at the subunit interface. In the inner leaflet, a buried lipid pose was found at the subunit interface, determined by interactions with T274 and W217 of the M3 and M1 helices, respectively. (**B**) In the open state, six lipid interaction sites were identified from simulations, out of which three sites were known from previous structures and one presumably occupied by a detergent molecule (*Hu et al., 2018*) (thin blue stripes). In the outer leaflet, an additional lipid site could be found at the subunit interface in the open state, interacting with P250 on the M2-M3 loop. Additionally, the site at the principal intrasubunit site is slightly closer to the M4 helix compared to the closed state, and the lipid site at the principal intrasubunit site has an increased likelihood of tail penetration into an allosteric pocket compared to the closed state. Lipid sites in the inner leaflet are at positions equivalent to those in the closed state.

of protein dynamics to lipid interactions. Indeed, the unrestrained atomistic MD simulations studied here were not expected to capture the maximal duration of stable contacts, as indicated by some interaction times approaching the full 1.7 μs trajectory (*Figure 2*). Nevertheless, simulations were of sufficient length to sample exchange of up to four lipids, particularly around the M4 helix. Calculation of lipid lateral diffusion coefficients resulted in average displacements at the end of simulations of 2.15 nm for lipids initially interacting with the protein surface, roughly corresponding to lipids diffusing out to the fourth lipid shell. Diffusion of bulk lipids was faster, allowing lipids originally 3.16 nm away from the protein surface to ingress the first lipid shell. This observation underscores the potential for lipid exchange events even among lipids initially distant from the protein surface. Of course, duration of exceptionally stable interactions, such as those involving T274 (*Figure 2C*), inevitably remain bounded by the length of our simulations. Still, diffusion metrics, supported by robust statistical analysis encompassing diverse starting conditions (500 trajectories), enable confident estimation of relative interaction times.

Sampling was further simplified by performing simulations in a uniform POPC membrane. Prior experiments have been conducted to assess the sensitivity of GLIC in varying lipid environments (*Labriola et al., 2013*; *Carswell et al., 2015*; *Menny et al., 2017*), indicating that GLIC remains fully functional in pure POPC bilayers. In our cryo-EM experiments, the protein was recombinantly expressed from *Escherichia coli*, which means that the experimental density would likely represent phosphatidylglycerol or phosphatidylethanolamine lipids. However, as the molecular identities of

bound lipids could not be precisely determined, POPC lipids were built for straightforward comparison with simulation poses. While it appears that GLIC is capable of gating in a pure POPC bilayer, it remains plausible that its function could be influenced by different lipid species, especially due to the presence of multiple charged residues around the TMD/ECD interface which might interact differently with different lipid head groups. Further experiments would be needed to confirm whether the state dependence observed in simulations is also lipid-dependent. It is possible that certain types of lipids bind in one but not the other state, or that certain states are stabilized by a particular lipid type.

In simulations, we were able to identify both long-lasting and highly occupied lipid sites by analyzing a large number of parallel trajectories (500 instances of each subunit). Simulation frames could further be classified into closed or open states by projection onto a previously published Markov state model of GLIC gating, which had been validated against electrophysiology recordings and structural features (*Bergh et al., 2021*). In particular, as simulations were initiated by placing the GLIC structures in generated POPC bilayers, it is reassuring that lipid binding sites identified in simulations appear to correspond closely to densities representing lipids bound tightly enough in the cryo-EM data to be retained during reconstitution (*Figure 1—figure supplement 1C, D*). The resulting occupancy plots enabled confident distinction between functional annotations while preserving the protein's flexibility and capacity to transition between states throughout the simulations. Interestingly, the occupational densities varied remarkably little between resting and activating conditions (*Figure 1—figure supplement 1*), indicating state rather than pH dependence in lipid interactions, also further justifying the approach of merging closed-state GLIC cryo-EM datasets collected at different pH conditions to resolve lipids.

The three lipids identified in the inner leaflet may play primarily structural roles, as they did not vary substantially between closed and open states (*Figure 5*). Principal-face and intersubunit sites in our closed cryo-EM structure were largely superimposable with those in open X-ray structures, and all three sites contained comparable densities in closed- and open-state simulation frames (*Figure 3A*, *Figure 1—figure supplement 1D*). A particularly stable contact in this leaflet featured deep intercalation of a lipid tail between channel subunits (*Figure 4—figure supplement 1*), as represented by lipid interactions persisting over 1 μs at residue T274, the longest-lasting in the entire protein (*Figure 2*). A lipid in this site was also reported in previous open-state X-ray structures of GLIC (*Bocquet et al., 2009*; *Hu et al., 2018*). Removal of the sidechain at W217 substantially abbreviated interactions with its buried neighbor T274 (*Figure 3*), suggesting this hydrophobic contact helps to stabilize lipid penetration. Notably, the same W217A substitution was previously shown to suppress GLIC function and was considered the most impactful of several aromatic residues on the transmembrane surface (*Therien and Baenziger, 2017*). Interestingly, a lack of double bonds in the intercalating tail was associated with the lipid head group interacting more closely with the protein surface, indicating a possible role for saturation in overall lipid-binding modes (*Figure 3*). The relevance of lipid intercalation and saturation has been difficult to establish on the basis of structural data, given that resolved lipid tails are often limited to a few hydrocarbons beyond the glycerol group. An important extension of this work may be to apply similar analyses to simulations of mixed-lipid systems, albeit at substantially higher computational cost.

Lipid interactions in the outer leaflet were more variable, both between functional states and experimental methods, possibly reflecting larger conformational changes of the protein in this region. At the complementary-face site - one of the most consistently documented in previous X-ray structures - simulations indicated that lipid tails could penetrate deeper into the helical bundle upon channel opening (*Figure 5B*), and even further in a state associated with desensitization (*Figure 4—figure supplement 2A*). Interestingly, buried portions of this site have been shown to host allosteric modulators such as propofol, substituting for lipid-tail interactions (*Heusser et al., 2018*). State dependence was also suggested at the principal-face site, where the lipid head preferred a pose tilted in toward M3 in the closed state, but tilted out toward M4 in the open state. However, our closed cryo-EM structure was more consistent with open-state simulations, and with a detergent built in an open X-ray structure (*Hu et al., 2018*). Indeed, the outward orientation was also sampled in closed-state simulations, albeit with lower frequency (*Figure 4—figure supplement 2D*). The relevance of the inward-facing pose, possibly in the context of an alternative lipid head group, remains to be determined. The most provocative indication of state dependence was at the outer subunit interface, which did not contain strong density in either cryo-EM or simulations data for the closed state (*Figure 5A*). In contrast,

lipid occupancy at this site was evident in open simulations (*Figure 5B*), including state-dependent contacts with P250, a conserved M2-M3 residue implicated in channel gating (*Kaczor et al., 2022*). Although lipids have yet to be resolved at this intersubunit site in GLIC, they have been reported in open-state structures of both ELIC (*Petroff et al., 2022*) and GluCl (*Althoff et al., 2014*), and in a desensitized neuronal GABA(A) receptor (*Laverty et al., 2019*). The state-dependent binding event at this site preceded pore opening in MSMs, where lipid binding coincided with crossing a smaller energy barrier between closed and intermediate states, followed by pore opening at the main energy barrier between intermediate and open states (*Figure 4—figure supplement 3D*). Further, since the P250-lipid interaction was characterized by relatively long residence times (*Figure 2*), it is possible this lipid interaction has a role to play in GLIC gating.

Finally, this work demonstrates the combined power of cryo-EM and MD simulations to characterize membrane-protein lipid interactions. In leveraging extensive MD simulations to build lipids for the first time into a closed-state structure of GLIC, we were also able to capture features of lipid-tail saturation and potential state dependence that were not available from structural data alone. Knowledge of such interactions may prove useful in the development of lipid-like drugs or lipid-focused treatments of diseases related to malfunction of pLGICs.

# Materials and methods
## MD simulations and analysis

We analyzed previously published MSMs of GLIC gating under both resting and activating conditions (*Bergh et al., 2021*). Resting conditions corresponded to pH 7, at which GLIC is nonconductive in functional experiments, with all acidic residues modeled as deprotonated. Activating conditions corresponded to pH 4.6, at which GLIC is conductive and has been crystallized in an open state (*Bocquet et al., 2009*). These conditions were modeled by protonating a group of acidic residues (E26, E35, E67, E75, E82, D86, D88, E177, E243; H277 doubly protonated) as previously described (*Nury et al., 2011*). In order to sample lipid movement more extensively, each of the 100 simulations was extended to 1.7 µs, resulting in an additional 50 µs sampling compared to *Bergh et al., 2021*. For state classification, the additional simulation frames were projected onto the MSMs trained on the previously published simulations.

Time-based measures of protein-lipid interactions, such as mean duration times and exchange of interactions, were calculated for the 100× 1.7-µs-long simulations using prolintpy (*Sejdiu and Tieleman, 2021*) with a 4 Å interaction cutoff. Analysis of lateral lipid diffusion in individual simulations was carried out for two disjoint sets of lipids: the first lipid shell defined as lipids with any part within 4 Å of the protein surface (~90 lipids), and bulk lipids consisting of all other lipids (~280 lipids). Mean square displacements of each lipid set were calculated using GROMACS 2021.5 (*Abraham et al., 2015*) with contributions from the protein center of mass removed. Diffusion coefficients for each set, $D_A$, were calculated using the Einstein relation (*Equation 1*) by estimating the slope of the linear curve fit to the data.

$$\lim_{t \to \infty} \langle \| \mathbf{r}_i\left(t\right) - \mathbf{r}_i\left(0\right) \|^2 \rangle_{i \in A} = 4 D_A t \qquad (1)$$

where $\mathbf{r}_i\left(t\right)$ is the coordinate of the center of mass of lipid $i$ of set $A$ at time $t$ and $D_A$ is the self-diffusion coefficient.

To investigate state-dependent protein-lipid interactions, all trajectory frames were classified according to the five clusters in *Bergh et al., 2021*, resulting in 19,555 and 21,941 frames for the closed and open protonated states, respectively, and 20,632 and 13,452 frames for the closed and open deprotonated states, respectively. Occupied densities were calculated from the frames corresponding to each cluster using VMD's volmap tool (*Humphrey et al., 1996*) for a selection of all whole lipids within 3 Å of the protein surface. In order to get representative snapshots of the protein-lipid interactions, we calculated global correlations between the computational densities and the selection of lipids used to derive the same densities using GROmaps (*Briones et al., 2019*) and extracted 100 frames with the highest correlations. To ensure that all frames would not originate from a single trajectory, a maximum of 40% of the highest-correlating frames were allowed to originate from a single trajectory.

To explore the effect of mutations, we ran sets of 8×1.7 µs simulations on open wild-type GLIC (PDB ID: 4HFI) (*Sauguet et al., 2013*) with the modification W217A, T274A, or T274W. Each mutated protein was independently embedded in a POPC lipid bilayer, solvated by water and 0.1 M NaCl. Acidic residues (E26, E35, E67, E75, E82, D86, D88, E177, E243; H277 doubly protonated) were protonated to approximate the probable pattern at pH 4.6, as previously described (*Nury et al., 2011*; *Bergh et al., 2021*). Constant pressure and temperature relaxation was carried out in four steps, restraining all heavy atoms in the first cycle, followed by only backbone atoms, then Cα atoms, and finally the M2 helices, for a total of 76 ns. Equilibration and production runs were performed using the Amber99SB-ILDN force field with Berger lipids (*Berger et al., 1997*), together with the TIP3P water model. Temperature coupling was achieved with velocity rescaling (*Bussi et al., 2007*) and pressure coupling with the Parrinello-Rahman barostat (*Parrinello and Rahman, 1981*). Open-state mutagenesis simulations were prepared and run with GROMACS versions 2018.4 and 2020.5 (*Abraham et al., 2015*). Simulations extended from those described in *Bergh et al., 2021*, were prepared in a similar way as described above but with GROMACS versions 2018.4 and 2019.3.

Conformational analysis and visualization of protein and lipids were performed with MDAnalysis (*Michaud-Agrawal et al., 2011*; *Gowers et al., 2016*), PyEMMA 2.5.7 (*Scherer et al., 2015*), and VMD (*Humphrey et al., 1996*).

## Cryo-EM sample preparation and data acquisition

Experiments used in this project were previously reported in *Rovšnik et al., 2021*. Briefly, C43(DE3) *E. coli* transformed with GLIC-MBP in vector pET-20b were cultured overnight at 37°C. Cells were then inoculated 1:100 into the 2xYT media containing 100 µg/mL ampicillin and grown at 37°C until they reached $OD_{600}$=0.7. Next, the cells were induced with 100 µM isopropyl-β-D-1-thiogalactopyranoside, and shaken overnight at 20°C. Membranes were harvested from cell pellets by sonication and ultra-centrifugation in buffer A (300 mM NaCl, 20 mM Tris-HCl pH 7.4) supplemented with 1 mg/mL lysozyme, 20 µg/mL DNase I, 5 mM $MgCl_2$, and protease inhibitors. At this point, the cells were either frozen or immediately solubilized in 2% *n*-dodecyl-β-D-maltoside (DDM). Amylose affinity resin (NEB) was used for purification of fusion protein in batch which was eluted in buffer B (buffer A with 0.02% DDM) with 2–20 mM maltose followed by size exclusion chromatography in buffer B. After overnight thrombin digestion, GLIC was isolated from its fusion partner by size exclusion in buffer B at pH 7, or in buffer B with citrate at pH 5 or 3 substituted for Tris. The purified protein was concentrated to 3–5 mg/mL by centrifugation.

Quantifoil 1.2/1.3 Cu 300 mesh grids (Quantifoil Micro Tools) were used for sample preparation. The grids were glow-discharged in methanol vapor directly before 3 µL of sample was applied to them. Following a 1.5 s blot they were plunge-frozen into liquid ethane using an FEI Vitrobot Mark IV. Movies were collected on an FEI Titan Krios 300 kV microscope with a K2-Summit direct electron detector camera at nominal ×165,000 magnification, equivalent to a pixel spacing of 0.82 Å. A total dose of 40.8 e⁻/Å² was used to collect 40 frames over 6 s, with defocus values ranging from –2.0 to –3.8 µm.

## Image processing

Processing was performed through the RELION 4.0-beta-2 pipeline (*Kimanius et al., 2021*). Data from three different grids, at pH 7, 5, and 3, were merged and processed together. Motion correction was performed with Relion's own implementation (*Zivanov et al., 2019*), followed by a defocus estimation from the motion-corrected micrographs using CtfFind4 (*Rohou and Grigorieff, 2015*). Following manual picking, initial 2D classification was performed to generate references for autopicking. Particles were extracted after autopicking and binned, followed by a 2D classification of the entire dataset. A smaller subset of particles was used to generate an initial model. All subsequent processing steps were done using fivefold symmetry. The initial model was used in a consecutive 3D auto-refinement and the acquired alignment parameters were used to identify and remove noisy particles through multiple rounds of pre-aligned 2D and 3D classification. The final set of particles was then refined, using the best 3D reconstruction as reference.

Particle polishing and per-particle CTF parameters were estimated from the resulting reconstruction using RELION 4.0-beta-2. Global beam-tilt was estimated from the micrographs and correction applied. Micelle density was subtracted and the final 3D auto-refinement was performed using a soft

mask covering the protein, followed by post-processing. Local resolution was estimated using the Relion implementation. Autosharpen map tool in PHENIX 1.19.2-4158 (*Adams et al., 2010*) was used to improve the visibility of peripheral lipid and detergent densities around the protein.

## Model building

The model was built from a template cryo-EM structure determined at pH 7 (PDB ID: 6ZGD; *Rovšnik et al., 2021*). A monomer of that model was fit to the reconstructed density and fivefold symmetry was applied with PHENIX 1.19.2-4158 through NCS restraints detected from the reconstructed cryo-EM map, to generate a complete channel. The model was incrementally adjusted in COOT 0.9.6 EL (*Emsley and Cowtan, 2004*) and re-refined until conventional quality metrics were optimized in agreement with the reconstruction. Model statistics are summarized in *Table 1*.

Lipids were manually built in COOT by importing a canonical SMILES format of POPC (*Kim et al., 2021*) and adjusting it individually into the cryo-EM density in each of the sites associated with a single subunit, based in part on visual inspection of lipid densities from simulations, as described above. After building, fivefold symmetry was applied to generate lipids at the same sites in the remaining four subunits.

Molecular visualizations were created with VMD (*Humphrey et al., 1996*) and ChimeraX (*Goddard et al., 2018*).

## Acknowledgements

The authors would like to thank the Swedish Cryo-EM National Facility staff, especially Marta Carroni and Stefan Fleischmann from Stockholm and Michael Hall from Umeå, for kind assistance with data collection. This work was supported by the Knut and Alice Wallenberg Foundation, and the Swedish Research Council (2019-04433, 2021-05806), the Swedish e-Science Research Centre (SeRC), the BioExcel Center of Excellence (EU-101093290), and the Sven and Lily Lawskis Fond (UR). Cryo-EM data were collected at the Swedish national cryo-EM facility funded by the Knut and Alice Wallenberg Foundation, Erling Persson, and Kempe Foundations. Computational resources were provided by the Swedish National Infrastructure for Computing (SNIC).

## Additional information

### Funding

| Funder | Grant reference number | Author |
| --- | --- | --- |
| Knut och Alice Wallenbergs Stiftelse | | Erik Lindahl |
| Vetenskapsrådet | 2019-04433 | Erik Lindahl |
| Sven och Lilly Lawskis Fond för Naturvetenskaplig Forskning | | Urška Rovšnik |
| Swedish e-Science Research Centre | | Erik Lindahl Rebecca Howard |
| European Commission | 101093290 | Erik Lindahl |
| Vetenskapsrådet | 2021-05806 | Erik Lindahl |

The funders had no role in study design, data collection and interpretation, or the decision to submit the work for publication.

### Author contributions

Cathrine Bergh, Urška Rovšnik, Conceptualization, Data curation, Investigation, Methodology, Writing – original draft, Writing – review and editing; Rebecca Howard, Supervision, Writing – original draft, Project administration, Writing – review and editing; Erik Lindahl, Supervision, Funding acquisition, Project administration, Writing – review and editing

## Author ORCIDs
Cathrine Bergh http://orcid.org/0000-0001-7540-5887
Erik Lindahl https://orcid.org/0000-0002-2734-2794

Reviewer #1 (Public Review): https://doi.org/10.7554/eLife.86016.3.sa1
Reviewer #2 (Public Review): https://doi.org/10.7554/eLife.86016.3.sa2
Author Response https://doi.org/10.7554/eLife.86016.3.sa3

## Additional files

### Supplementary files
• MDAR checklist

### Data availability
Cryo-EM density maps of the pentameric ligand-gated ion channel GLIC in detergent micelles have been deposited in the Electron Microscopy Data Bank under accession number EMD-15649. The deposition includes the cryo-EM sharpened and unsharpened maps, both half-maps and the mask used for final FSC calculation. Coordinates of the model have been deposited in the Protein Data Bank under accession number 8ATG. State-clustered simulation frames, computational lipid densities, parameter files, and full trajectories of mutant simulations as reported in this work can be accessed at https://doi.org/10.5281/zenodo.7058272. Previously published computational models and simulation files can be accessed at https://doi.org/10.5281/zenodo.5500174.

The following datasets were generated:

| Author(s) | Year | Dataset title | Dataset URL | Database and Identifier |
| --- | --- | --- | --- | --- |
| Bergh C, Rovsnik U, Howard R, Lindahl E | 2022 | Discovery of lipid binding sites in a ligand-gated ion channel by integrating simulations and cryo-EM | https://zenodo.org/record/7058272 | Zenodo, 10.5281/zenodo.7058272 |
| Bergh C, Rovsnik U, Howard R, Lindahl E | 2022 | Pentameric ligand-gated ion channel GLIC with bound lipids | https://www.ebi.ac.uk/emdb/EMD-15649 | EMDB, EMD-15649 |
| Bergh C, Rovsnik U, Howard R, Lindahl E | 2022 | Pentameric ligand-gated ion channel GLIC with bound lipids | https://www.rcsb.org/structure/8ATG | RCSB Protein Data Bank, 8ATG |

The following previously published dataset was used:

| Author(s) | Year | Dataset title | Dataset URL | Database and Identifier |
| --- | --- | --- | --- | --- |
| Bergh C, Heusser S, Howard R, Lindahl E | 2021 | Markov State Models of Proton- and Gate-Dependent Activation in a Pentameric Ligand-Gated Ion Channel | https://doi.org/10.5281/zenodo.5500174 | Zenodo, 10.5281/zenodo.5500174 |

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
