## [Editor Report · eLife assessment]

The authors use a combination of structural and MD simulation approaches to characterize phospholipid interactions with the pentameric ligand-gated ion channel, GLIC. The general agreement between structures and simulations increases confidence in the description of the lipid interaction poses and provides a **solid** basis for the prediction of a state-dependent interaction site where lipids could dynamically modulate channel gating. The results will be very **useful** to understand the nature of phospholipid interactions with pentameric ligand-gated ion channels, although the functional or structural significance of these lipid interactions remains to be verified.

---

## [Referee Report · Reviewer #1 (Public Review)]

The authors use a combination of structural and MD simulation approaches to characterize phospholipid interactions with the pentameric ligand-gated ion channel, GLIC. By analyzing the MD simulation data using clusters of closed and open states derived previously, the authors also seek to compare lipid interactions between putative functional states. The ultimate goal of this work is to understand how lipids shape the structure and function of this channel.

The strengths of this article include the following:

1. The MD simulation data provide extensive sampling of lipid interactions in GLIC, and these interactions were characterized in putative closed and open states of the channel. The extensive sampling permits confident delineation of 5-6 phospholipid interaction sites per subunit. The agreement in phospholipid binding poses between structures and the all-atom MD simulations supports the utility of MD simulations to examine lipid interactions.

2. The study presents phospholipid binding sites/poses that agree with functionally important lipid binding sites in other pLGICs, supporting the notion that these sites are conserved. For example, the authors identify interactions of POPC at an outer leaflet intersubunit site that is specific for the open state. This result is quite interesting as phospholipids or drugs that positively modulate other pLGICs are known to occupy this site. Also, the effect of mutating W217 in the inner leaflet intersubunit site suggests that this residue, which is highly conserved in pLGICs, is an important determinant of the strength of phospholipid interactions at this site. This residue has been shown to interact with phospholipids in other pLGICs and forms the binding site of potentiating neurosteroids in the GABA(A) receptor.

Comments on the revised version:

We appreciate the authors' thorough response and revisions.

Specifically, the authors address the issue of interaction times by providing measures of the diffusion coefficients and mean displacements of the lipids. These show that there is sufficient movement of lipids within the first shell to indicate that certain residues are forming binding interactions with lipids while others are not. Longer simulation times would be necessary to determine the strength of these interactions and how they may differ between different conformations.

---

## [Referee Report · Reviewer #2 (Public Review)]

The authors convincingly show multiple inner and outer leaflet non-protein (lipid) densities in a cryo-EM closed state structure of GLIC, a prokaryotic homologue of canonical pentameric ligand-gated ion channels, and observe lipids in similar sites during extensive simulations at both resting and activating pH. The simulations not only corroborate structural observations but also suggest the existence of a state-dependent lipid intersubunit site only occupied in the open state. These important findings will be of considerable interest to the ion channel community and provide new hypotheses about lipid interactions in conjunction with channel gating.

Comments on the revised version:

The authors have addressed all of my comments.

---

## [Author Response]

The following is the authors’ response to the original reviews.

Thank you for the thoughtful consideration of our work, including both reviewers’ constructive comments. Our apologies for taking some extra time for this revision, but we wanted to adress comments thoroughly with new analyses, not to mention a PhD defense, parental leave and my teaching ultimately being the bottleneck for the team’s work!

**Reviewer #1 (Public Review):**
The authors use a combination of structural and MD simulation approaches to characterize phospholipid interactions with the pentameric ligand-gated ion channel, GLIC. By analyzing the MD simulation data using clusters of closed and open states derived previously, the authors also seek to compare lipid interactions between putative functional states. The ultimate goal of this work is to understand how lipids shape the structure and function of this channel.The strengths of this article include the following:1. The MD simulation data provide extensive sampling of lipid interactions in GLIC, and these interactions were characterized in putative closed and open states of the channel. The extensive sampling permits confident delineation of 5-6 phospholipid interaction sites per subunit. The agreement in phospholipid binding poses between structures and the all-atom MD simulations supports the utility of MD simulations to examine lipid interactions.1. The study presents phospholipid binding sites/poses that agree with functionally-important lipid binding sites in other pLGICs, supporting the notion that these sites are conserved. For example, the authors identify interactions of POPC at an outer leaflet intersubunit site that is specific for the open state. This result is quite interesting as phospholipids or drugs that positively modulate other pLGICs are known to occupy this site. Also, the effect of mutating W217 in the inner leaflet intersubunit site suggests that this residue, which is highly conserved in pLGICs, is an important determinant of the strength of phospholipid interactions at this site. This residue has been shown to interact with phospholipids in other pLGICs and forms the binding site of potentiating neurosteroids in the GABA(A) receptor.Weaknesses of this article include the following:1. The authors describe in detail state-dependent lipid interactions from the MD simulations; however, the functional significance of these findings is unclear. GLIC function appears to be insensitive to lipids, although this understanding is based on experiments where GLIC proteoliposomes were fused to oocyte membranes, which may not be optimal to control the lipid environment. Without functional studies of GLIC in model membranes, the lipid dependence of GLIC function is not definitively known. Therefore, it is difficult to interpret the meaning of these state-dependent lipid interactions in GLIC.1. It is unlikely that the bound phospholipids in the GLIC structures, which are co-purified from *E. coli* membranes, are POPC. Rather, these are most like PE or PG lipids. While it is difficult to accommodate mixed phospholipid membranes in all-atom MD simulations, the choice of POPC for this model, while practically convenient, seems suboptimal, especially since it is not known if PE or PG lipids modulate GLIC function. Nevertheless, it is striking that the overall binding poses of POPC from the simulations agree with those identified in the structures. It is possible that the identity of the phospholipid headgroup will have more of an impact on the strength of interactions with GLIC rather than the interaction poses (see next point).1. The all-atom MD simulations provide limited insight into the strength of the POPC interactions at each site, which is important to interpret the significance of these interactions. It is unlikely that the system has equilibrated within the 1.7 microseconds of simulation for each replicate preventing a meaningful assessment of the lipid interaction times. Although the authors report exchange of up to 4 POPC interacting at certain residues in M4, this may not represent binding/unbinding events (depending on how binding/interaction is defined), since the 4 Å cutoff distance for lipid interactions is relatively small. This may instead be a result of small movements of POPC in and out of this cutoff. The ability to assess interaction times may have been strengthened if the authors performed a single extended replicate up to, for example, 10-20 microseconds instead of extending multiple replicates to 1.7 microseconds.
**Reviewer #2 (Public Review):**
The authors convincingly show multiple inner and outer leaflet non-protein (lipid) densities in a cryo-EM closed state structure of GLIC, a prokaryotic homologue of canonical pentameric ligand-gated ion channels, and observe lipids in similar sites during extensive simulations at both resting and activating pH. The simulations not only corroborate structural observations, but also suggest the existence of a state-dependent lipid intersubunit site only occupied in the open state. These important findings will be of considerable interest to the ion channel community and provide new hypotheses about lipid interactions in conjunction with channel gating.

Recommendations for the authors: please note that you control which, if any, revisions, to undertake

In particular, a discussion of whether the timescale of the simulations permit measurements of residence or interaction times of the lipids should be addressed.
**Reviewer #1 (Recommendations for the authors):**
Comment 1.1: The authors may consider expanding the discussion about the significance of state-dependent lipid interactions. On the one hand, they emphasize state-dependent interactions of POPC with closed and open states in the outer leaflet in the results. On the other hand, they state that GLIC is insensitive to its lipid environment. What is the significance of the state-dependent interactions of POPC in GLIC, if any? It is possible that GLIC agonist responses are sensitive to phospholipids (such as PE or PG found in *E. coli*)? The state-dependent differences in lipid interaction identified in this study support this possibility and suggest the need to better understand the effects of phospholipids on GLIC function.

Response 1.1: We agree with the reviewer that this is an interesting question and we have therefore extended the discussion with additional references on the functional effects on GLIC of various lipid membranes:

p. 11 (Discussion)

“Sampling was further simplified by performing simulations in a uniform POPC membrane. Prior experiments have been conducted to assess the sensitivity of GLIC in varying lipid environments (Labriola et al., 2013; Carswell et al., 2015; Menny et al., 2017), indicating that GLIC remains fully functional in pure POPC bilayers. In our cryo-EM experiments, the protein was recombinantly expressed from *E. coli*, which means that the experimental density would likely represent phosphatidylglycerol or phosphatidylethanolamine lipids.However, as the molecular identities of bound lipids could not be precisely determined, POPC lipids were built for straightforward comparison with simulation poses. While it appears that GLIC is capable of gating in a pure POPC bilayer, it remains plausible that its function could be influenced by different lipid species, especially due to the presence of multiple charged residues around the TMD/ECD interface which might interact differently with different lipid head groups. Further experiments would be needed to confirm whether the state dependence observed in simulations is also lipid-dependent. It is possible that certain types of lipids bind in one but not the other state, or that certain states are stabilized by a particular lipid type.”

Comment 1.2: It would be helpful to state in the discussion that the co-purified lipids from GLIC structures are likely PE or PG from *E. coli* membranes. Nevertheless, it is interesting that the phospholipid poses from the structures generally agree with those identified from the MD simulations using PC.

Response 1.2: Good point. We have clarified in the discussion that the native lipids in the cryo-EM structure are likely PG or PE lipids, as quoted in the preceding Response.

Comment 1.3: The authors describe a more deeply penetrating interaction of POPC in the outer intrasubunit cleft in the open state, but this is difficult to appreciate from the images in Fig. 4B, 4E or S3B. The same is true of the deep POPC interaction at the outer intersubunit site. It may be helpful to show these densities from a different perspective to appreciate the depth of these binding poses.

Response 1.3: We have added Figure 4 – figure supplement 1 to better show the depth of lipid binding poses, especially the ones in the outer leaflet intrasubunit cleft and at the inner intersubunit site, and cited the figure on p. 7 (Results).

Comment 1.4: The representation of the lipid densities in Fig. 4B is not easy to interpret. First, the meaning of resting versus activating conditions and closed versus open states can be easily missed for readers who are not familiar with the author's previous study. It may be helpful to describe this (i.e. how open and closed state clusters were generated from structures determined in resting and activating conditions) in greater detail in either the figure legend, results or methods. Second, the authors state that there are differences in lipid poses between the closed and open states but not resting and activating conditions. With the exception of the intersubunit density, this is difficult to appreciate from Fig. 4B. As stated in point #3, the difference, for example, in the complementary intrasubunit site may be better appreciated with an image from a different perspective.

Response 1.4: Acknowledged - the distinction between resting and activating conditionsv.s. open and closed states can be confusing. We have tried to clarify these differences at the beginning of the results section, the methods section, and in the caption of Figure 4. Regarding differences in lipid poses between open and closed states, we agree it is difficult to appreciate from Figure 4, but here we refer the reader to Figure 4 – figure supplement 2 for an overlay between open and closed densities. Additionally, we now added Figure 1 – figure supplement 1 which provides lipid densities for all five subunits and overlays with the build cryo-EM lipids, possibly making differences easier to appreciate. Regarding images from different perspectives, we trust the new figure supplement described in Response 1.3 provides a better perspective.

p. 3 (Results)

“For computational quantification of lipid interactions and binding sites, we used molecular simulations of GLIC conducted under either resting or activating conditions (Bergh et al., 2021a). As described in Methods, resting conditions corresponded to neutral pH with most acidic residues deprotonated; activating conditions corresponded to acidic pH with several acidic residues protonated. Both open and closed conformations were present in both conditions, albeit with different probabilities.”

p. 8 (Figure 4)

“Overlaid densities for each state represent simulations conducted under resting (dark shades) or activating (light shades) conditions, which were largely superimposable within each state.”

p. 24 (Methods)

“We analyzed previously published MSMs of GLIC gating under both resting and activating conditions (Bergh et al., 2021a). Resting conditions corresponded to pH 7, at which GLIC is nonconductive in functional experiments, with all acidic residues modeled as deprotonated. Activating conditions corresponded to pH 4.6, at which GLIC is conductive and has been crystallized in an open state (Bocquet et al., 2009). These conditions were modeled by protonating a group of acidic residues (E26, E35, E67, E75, E82, D86, D88, E177, E243; H277 doubly protonated) as previously described (Nury et al., 2011).”

Comment 1.5: The new closed GLIC structure was obtained by merging multiple datasets. What were the conditions of the datasets used? Was it taken from samples in resting or also activating conditions?

Response 1.5: We have updated the Results, Discussion, and Methods to clarify this important point, in particular by merging datasets and rerunning the classification:

p. 3 (Results)

“In our cryo-EM work, a new GLIC reconstruction was generated by merging previously reported datasets collected at pH 7, 5, and 3 (Rovšnik et al., 2021). The predominant class from the merged data corresponded to an apparently closed channel at an overall resolution of 2.9 Å, the highest resolution yet reported for GLIC in this state (Figure 1 – figure supplement 2, Table 1).”

p. 11 (Discussion)

“Interestingly, the occupational densities varied remarkably little between resting and activating conditions (Figure 1 – figure supplement 1), indicating state- rather than pH- dependence in lipid interactions, also further justifying the approach of merging closed- state GLIC cryo-EM datasets collected at different pH conditions to resolve lipids.”

p. 14 (Methods)

“After overnight thrombin digestion, GLIC was isolated from its fusion partner by size exclusion in buffer B at pH 7, or in buffer B with citrate at pH 5 or 3 substituted for Tris. The purified protein was concentrated to 3–5 mg/mL by centrifugation.[...] Data from three different grids, at pH 7, 5, and 3, were merged and processed together.”

Comment 1.6: In Fig. 3D, do the spheres represent the double bond? If so, please state in the legend

Response 1.6: We have clarified in the legend of Figure 3D that the yellow spheres on the lipid tails represent a double bond.

Comment 1.7: In Fig. 3E, what is the scale of the color representation?

Response 1.7: We have clarified in the legend of Figure 3E that colors span 0 (white) to 137015 contacts (dark red).

**Reviewer #2 (Recommendations For The Authors):**
Comment 2.1: I'm not sure I fully understand how the final lipids were modeled (built). Fig. 1 caption suggests they may have been manually built? I understand that the idea was to place them in the overlap of simulation densities and structure densities, but can the authors please clarify if there were any quantifiable conditions that were employed during this process or if this was entirely manual placement in a pose that looked good? Regardless, it would be helpful to see an overlay of the built lipids with both the cryo and simulation densities (e.g., overly of Fig. 1F/H and G/H) to better visualize how the final built lipids compare.

Response 2.1: We thank the reviewer for pointing out unclarities regarding our methods. We have extended the methods section to clarify how the lipids were manually built in the cryo-EM structure. We have also added Figure 1 – figure supplement 1 showing overlays of the computational densities and built cryo-EM lipids.

p. 15 (Methods)

“Lipids were manually built in COOT by importing a canonical SMILES format of POPC (Kim et al., 2021) and adjusting it individually into the cryo-EM density in each of the sites associated with a single subunit, based in part on visual inspection of lipid densities from simulations, as described above. After building, 5-fold symmetry was applied to generate lipids at the same sites in the remaining four subunits.”

Comment 2.2: Regarding the state-dependent lipid entry to the outer leaflet intersubunit site associated with channel opening, if the authors could include a movie depicting this process that would be great. The current short explanation does not do this justice. Also, what were the dynamics of this process? Beyond the correlation between site occupancy and the pore being open, how did the timing of lipid entry/exit and pore opening/closing correlate?

Response 2.2: The point regarding the timing of state-dependent lipid binding at the subunit interface and pore opening is indeed an interesting one. We have added Figure 4 – figure supplement 3D showing that the state-dependent P250 lipid interaction precedes pore opening, as quantified by pore hydration levels, indicating a potential role in gating. The interaction between lipid binding and conformational change of the protein is also depicted in the newly added Figure 4 - video supplement 1, which we hope will be able to better communicate the conclusions regarding state-dependent interactions. We have also expanded the results and discussion to better explain these results:

p. 9 (Results)

“The lipid head made particularly close contacts with residue P250 on the M2-M3 loop, which undergoes substantial conformational change away from the pore upon channel opening, along with outer-leaflet regions of M1–M3 (Figure 4E, Figure 4—figure Supplement 3A,B,C, Figure 4—video 1). These conformational changes were accompanied by a flip of M1 residue F195, which blocked the site in the closed state but rotated inward to allow closer lipid interactions in the open state (Figure 4—figure Supplement 3C, Figure 4—video 1). Indeed, P250 was predominantly located within 3 Å of the nearest lipid atom in open- but not closed-state frames (Figure 4F). Despite being restricted to the open state, interactions with P250 were among the longest duration in all simulations (Figure 2C) and as these binding events preceded pore opening, it is plausible to infer a role for this state-dependent lipid interaction in the gating process (Figure 4 – figure supplement 3D).”

p. 12 (Discussion)

“The state-dependent binding event at this site preceded pore opening in MSMs, where lipid binding coincided with crossing a smaller energy barrier between closed and intermediate states, followed by pore opening at the main energy barrier between intermediate and open states (Figure 4 – figure supplement 3D). Further, since the P250- lipid interaction was characterized by relatively long residence times (Figure 2), it is possible this lipid interaction has a role to play in GLIC gating.”

Comment 2.3: Although the interaction times are helpful, I didn't get a great sense of how mobile the lipids are during the simulations. Can the authors discuss this a bit more. For example, are interaction times dominated by lipids that jiggle a bit away from a residue and then back again, vs how often are lipids exchanging with other lipids initially further away from the protein?

Response 2.3: We have now added various measures of lipid diffusion, both for initially interacting lipids and for bulk lipids, which are summarized in the new Figure 2 – figure supplement 1. We have further addressed the question of simulation timescales in Results, Discussion, and Methods. These numbers highlight that it is possible for lipids several nanometers away from the protein surface to exchange with lipids of the first lipid shell.

p. 3,6 (Results)

“Lateral lipid diffusion coefficients were estimated to 1.47 nm2/µs for bulk lipids and 0.68 nm2/µs for lipids of the first lipid shell (Figure 2 – figure supplement 1A), which is relatively slow compared to the timescales of each trajectory (1.7 µs). However, multiple residues throughout the M1, M3, and M4 helices exchanged contacts with 2-4 different lipid molecules in individual simulations (Figure 2C). Furthermore, 1.7-µs root mean square displacement of lipids originally in the first lipid shell was 2.15 nm, and 3.16 nm in the bulk bilayer, indicating such exchanges are not limited to nearby lipids (Figure 2 – figure supplement 1B). Thus, exchange events and diffusion estimates indicate that the duration of lipid contacts observed in this work can be at least partly attributed to interaction stabilities and not solely to sampling limitations.”

p. 11 (Discussion)

“Indeed, the unrestrained atomistic MD simulations studied here were not expected to capture the maximal duration of stable contacts, as indicated by some interaction times approaching the full 1.7-µs trajectory (Figure 2}). Nevertheless, simulations were of sufficient length to sample exchange of up to four lipids, particularly around the M4 helix. Calculation of lipid lateral diffusion coefficients resulted in average displacements at the end of simulations of 2.15 nm for lipids initially interacting with the protein surface, roughly corresponding to lipids diffusing out to the 4th lipid shell. Diffusion of bulk lipids was faster, allowing lipids originally 3.16 nm away from the protein surface to ingress the first lipid shell. This observation underscores the potential for lipid exchange events even among lipids initially distant from the protein surface. Of course, duration of exceptionally stable interactions, such as those involving T274 (Figure 2C), inevitably remain bounded by the length of our simulations. Still, diffusion metrics, supported by robust statistical analysis encompassing diverse starting conditions (500 trajectories), enable confident estimation of relative interaction times.“

p. 13 (Methods)

“Time-based measures of protein-lipid interactions, such as mean duration times and exchange of interactions, were calculated for the 100 x 1.7 µs-long simulations using prolintpy (Sejdiu and Tieleman, 2021) with a 4 Å interaction cutoff. Analysis of lateral lipid diffusion in individual simulations was carried out for two disjoint sets of lipids: the first lipid shell defined as lipids with any part within 4 Å of the protein surface (~90 lipids), and bulk lipids consisting of all other lipids (~280 lipids). Mean square displacements of each lipid set were calculated using GROMACS 2021.5 (Abraham et al., 2015b) with contributions from the protein center of mass removed. Diffusion coefficients for each set, DA, were calculated using the Einstein relation (Equation 1) by estimating the slope of the linear curve fit to the data.limt→∞⟨‖ri(t)−ri(0)‖2⟩i∈A=4DAt

where ri(t) is the coordinate of the center of mass of lipid i of set A at time t and DA is the self-diffusion coefficient.”

Comment 2.4: How symmetric or asymmetric are the cryo and simulation densities across subunits and was there subunit asymmetry in the final build lipids? I could not tell from any of the figures beyond the casual observation that they maybe look somewhat similar in Fig. 1?

Response 2.4: We thank the reviewer for this useful remark. We have clarified in the methods that the cryo-EM lipids were built in C5-symmetry, and thus the positions are symmetric. The computational densities were calculated independently for each subunit and are thus not necessarily symmetric. We have added Figure 1 – figure supplement 1 showing densities for all five subunits, also serving as an indication of convergence of the results.

p. 3 (Results)“Although the stochastic nature of simulations resulted in nonidentical lipid densities associated with the five GLIC subunits, patterns of lipid association were notably symmetric (Figure 1 – figure supplement 1).”

p. 14-15 (Methods)

“A smaller subset of particles was used to generate an initial model. All subsequent processing steps were done using 5-fold symmetry.[…] A monomer of that model was fit to the reconstructed density and 5-fold symmetry was applied with PHENIX 1.19.2-4158 through NCS restraints detected from the reconstructed cryo-EM map, to generate a complete channel.[…] After building, 5-fold symmetry was applied to generate lipids at the same sites in the remaining four subunits.”

Minor comments:Comment 2.5: Fig. 1 is probably not easy to follow for the general reader and the caption is very brief. I suggest adding an additional explanation to the caption and/or additional annotations to the figure to help a general reader step through this.

Response 2.5: We have expanded the caption of Figure 1 and clarified the meanings of colors, labels, and annotations.

Comment 2.6: Fig. 1B - Caption is confusing. I would not call the state separation lines outlines as they are not closed loops. Also, I see red/orange and two shades of blue whereas the caption mentions orange and blue only. The caption should also explicitly say what the black lines are (other cluster separations).

Response 2.6: We have edited the caption to better describe colors, annotations, and the meaning of the data:

p. 4 (Figure 1)

“(B) Markov state models were used to cluster simulations conducted under resting (R) or activating (A) conditions into five states, including closed (left of the light or dark orange lines) and open (right of the light or dark blue lines). Black lines mark edges of other state clusters derived from MSM eigenvectors. Experimental structures are highlighted as white circles.”

Comment 2.7: Fig. 3F caption appears to conflict with data where interaction with W217A appears longer than W217. I think the authors want to suggest here that W217A reduces contact time with T274 as stated in the main text.

Response 2.7: We have clarified in this legend that “Mutation of residue W217, lining this pocket, reveals shortened interactions at the T274 binding site” (p. 6, Figure 3).

Comment 2.8: Ref 25 and 26 are the same.

Response 2.8: Apologies; this mistake has been corrected.